# Protocol for Cell Colonization and Comprehensive Monitoring of Osteogenic Differentiation in 3D Scaffolds Using Biochemical Assays and Multiphoton Imaging

**DOI:** 10.3390/ijms24032999

**Published:** 2023-02-03

**Authors:** Kai Peter Sommer, Adrian Krolinski, Mohammad Mirkhalaf, Hala Zreiqat, Oliver Friedrich, Martin Vielreicher

**Affiliations:** 1Institute of Medical Biotechnology, Department of Chemical and Biological Engineering, Friedrich-Alexander University of Erlangen-Nürnberg, D-91052 Erlangen, Germany; 2School of Mechanical, Medical and Process Engineering, Queensland University of Technology, 2 George St., Brisbane, QLD 4000, Australia; 3Biomaterials and Tissue Engineering Research Unit, School of Aerospace, Mechanical and Mechatronic Engineering, University of Sydney, Sydney, NSW 2006, Australia

**Keywords:** bone cells, 3D-printed scaffolds, biograft, osteogenesis, collagen-I, mineralization, biochemical assays, imaging

## Abstract

The goal of bone tissue engineering is to build artificial bone tissue with properties that closely resemble human bone and thereby support the optimal integration of the constructs (biografts) into the body. The development of tissues in 3D scaffolds includes several complex steps that need to be optimized and monitored. In particular, cell–material interaction during seeding, cell proliferation and cell differentiation within the scaffold pores play a key role. In this work, we seeded two types of 3D-printed scaffolds with pre-osteoblastic MC3T3-E1 cells, proliferated and differentiated the cells, before testing and adapting different assays and imaging methods to monitor these processes. Alpha-TCP/HA (α-TCP with low calcium hydroxyapatite) and baghdadite (Ca_3_ZrSi_2_O_9_) scaffolds were used, which had comparable porosity (~50%) and pore sizes (~300–400 µm). Cell adhesion to both scaffolds showed ~95% seeding efficiency. Cell proliferation tests provided characteristic progression curves over time and increased values for α-TCP/HA. Transmitted light imaging displayed a homogeneous population of scaffold pores and allowed us to track their opening state for the supply of the inner scaffold regions by diffusion. Fluorescence labeling enabled us to image the arrangement and morphology of the cells within the pores. During three weeks of osteogenesis, ALP activity increased sharply in both scaffolds, but was again markedly increased in α-TCP/HA scaffolds. Multiphoton SHG and autofluorescence imaging were used to investigate the distribution, morphology, and arrangement of cells; collagen-I fiber networks; and hydroxyapatite crystals. The collagen-I networks became denser and more structured during osteogenic differentiation and appeared comparable in both scaffolds. However, imaging of the HA crystals showed a different morphology between the two scaffolds and appeared to arrange in the α-TCP/HA scaffolds along collagen-I fibers. ALP activity and SHG imaging indicated a pronounced osteo-inductive effect of baghdadite. This study describes a series of methods, in particular multiphoton imaging and complementary biochemical assays, to validly measure and track the development of bone tissue in 3D scaffolds. The results contribute to the understanding of cell colonization, growth, and differentiation, emphasizing the importance of optimal media supply of the inner scaffold regions.

## 1. Introduction

The accelerated growth of the world population combined with longer individual life expectancy (i.e., an increasing number of elderly people) [1] brings medical challenges such as the increasing prevalence of individuals with bone defects. For example, the number of individuals at high risk of fracture will approximately double between 2010 and 2040 [2,3]. Under certain boundary conditions, bone can regenerate completely, but there are also cases of impaired bone regeneration (e.g., due to avascular necrosis and osteoporosis) that are more common among the elderly [4,5]. Furthermore, large bone-size defects can exceed the natural regenerative capacity of bone; hence, bone grafts, usually auto- or allografts, must be used [6]. Autografts are the current gold standard with the best recuperative results [7]. Though the use of autografts, the likelihood of immune reactions and transmission of infections is minimal, but their availability is limited [4,8]. Allografts are more readily available, but offer reduced recuperative outcomes and increased risk of tissue rejection and infection transmission compared to autografts [4].

Bone tissue has a highly organized ECM (extracellular matrix) composed of organic and inorganic components and exhibits complex signaling cascades [9]. Bone tissue engineering (TE) aims to generate artificial bone tissue with a combination of scaffolds and cells [9,10,11,12,13]. Three-dimensional scaffolds act as a temporal ECM for cells and could be fabricated, for example, from inorganic materials, such as ceramics or ceramic composites [14,15]. The cells used for these purposes are commonly mesenchymal stem cells (MSCs) or pre-osteoblasts (e.g., MC3T3-E1 cells) [16,17,18]. The cells are seeded within the scaffold, where they stably adhere, followed by their proliferation and maturation. For osteogenic differentiation towards osteoblasts [13], cells are cultivated for at least three weeks, for instance, in an osteogenic medium. The medium’s osteogenic factors induce the expression and regulation of specific genes, act as co-factors for collagen-I fiber formation, and serve as a phosphate source for the formation of hydroxyapatite (HA) [19,20]. During the osteogenic differentiation of MC3T3-E1 cells, bone-related proteins are regulated in a temporal manner during the successive developmental stages, including proliferation (days 4–10), bone matrix formation/maturation (days 10–16), and mineralization (days 16–30) [12]. In the development of mesenchymal stem cells to pre-osteoblasts, they proliferate without the secretion of ECM. Later during their maturation, they start synthesizing fibrillar collagen-I and show an increasing level of alkaline phosphatase (ALP) activity. At the final stages, osteoid is synthesized, which becomes mineralized by the formation of HA. Throughout differentiation, the proliferation of cells decreases [21,22,23].

Certain aspects need to be considered when culturing cells in 3D scaffolds. These involve the exchange of nutrients, oxygen, and waste products between the scaffold core and medium. If the supply is insufficient (>1–2 mm aggregate thickness with only diffusion), hypoxic conditions are formed and cell necrosis and cell death arise. Consequently, the scaffold’s inner geometry must warrant the supply of nutrients to the scaffold core [24,25]. Scaffold porosity must be high enough for the medium to diffuse freely through the scaffolds’ interior. To prevent pore overgrowth, a balance should be established between the empty pore volume and the pore volume occupied by cells. This can be controlled by using scaffolds with suitable average pore sizes [26]. The initial cell population in the pores has a decisive influence on the further fate of the artificial tissue. Therefore, great attention must be paid to optimized cell seeding with high efficiency of adhesion of a suitable number of cells and homogeneous cell distribution in the porous network of the scaffold [27].

For developing suitable readouts for cell seeding, cell proliferation, and cell differentiation, two different well-established, 3D-printed, inorganic scaffold types were used: (a) scaffolds made from calcium-deficient hydroxyapatite with α-tricalcium phosphate (α-TCP/HA), and (b) scaffolds made from baghdadite. Both biomaterials are cyto-compatible and biodegradable with different kinetics [28] and the defined method for the 3D printing of α-TCP/HA (cement) and baghdadite (ceramic) has been demonstrated in previous studies [29,30]. α-TCP/HA consisting of HA and α-TCP is a biomaterial known to be osteoconductive and highly bioresorbable [29,30]. Studies have shown that HA induces the expression of osteospecific genes, such as ALP and collagen-I, while α-TCP increases bone-like tissue formation [31,32]. Baghdadite (Ca_3_ZrSi_2_O_9_) offers a good alternative to α-TCP/HA because of its superb bioactivity and mechanical properties [29,33]. The incorporation of Zr^2+^ into the calcium silicate results in increased mechanical stability and better biological properties including higher cell proliferation and expression of bone-related genes [28,34,35,36,37]. Highly porous baghdadite scaffolds were successfully used in several in vivo studies [6,38]. Both scaffold types exhibited 50% porosity and an average pore size in the range of ~300–400 µm, which is regarded as suitable for growing cell-seeded constructs (biografts) [24,25,39].

MC3T3-E1 pre-osteoblasts were used in all experiments. As mentioned, this cell line had been widely used for studies on osteogenesis and bone matrix formation [40,41,42]. Various tests and assays (with respect to cell adhesion, cell proliferation, and pore filling with cells) were applied to α-TCP/HA and baghdadite scaffolds, optimized, and their limitations were investigated. Furthermore, cell distribution and cell morphology as well as the formation of collagen-I fibers and HA during the course of osteogenic differentiation in the porous network were investigated by fluorescence and multiphoton imaging. In addition, ALP activity was investigated enzymatically over three weeks after switching to an osteogenic medium. The two scaffold types were compared with these assays.

The aim of this study was to optimize methods for the determination of a broad range of different parameters and to better understand the limitations and challenges involved. The hypothesis was that it was possible to measure and visualize, track over time, and optimize osteogenesis in 3D scaffolds using appropriate methods and assays. We used a multimodal approach to obtain maximum information on cell adhesion, cell distribution, and cell morphology in combination with the monitoring of selected cellular and ECM markers of osteogenesis. Particular attention was directed to the study of the 3D characteristics within the scaffolds. The results showed that the scaffold type needs to be considered when interpreting the findings. We also found that the pore colonization by cells largely depends on the internal geometry of the scaffolds. The assays for quantifying cell seeding, cell proliferation, and ALP activity are relatively straight forward. Additionally, imaging provided spatially resolved information on the distribution and morphology of cells, collagen-I fiber networks (SHG imaging), and hydroxyapatite crystals and was a very good complement to the biochemical assays. However, imaging data are, in part, more difficult to quantify. The multiphoton microscopy (MPM) method continuously provided non-invasive and depth-resolved 3D imaging [43,44,45] of critical tissue structures over time, and was particularly valuable for assessing the quality of the formed artificial tissues.

## 2. Results

Initial seeding and cultivation experiments were performed with human MSCs, and various baghdadite scaffold geometries, pore sizes, and lattice structures were initially tested. These tests included scaffold pretreatment as well as the determination of cell-seeding and cell-proliferation properties. Even though the seeding experiments were successful, we later observed insufficient proliferation rates with MSCs, which led to our abandonment of these experiments. As MSCs are known to already begin differentiating after around three weeks of culture [16], and cells produce only limited amounts of collagen-I for initial cell attachment, we switched to the MC3T3-E1 pre-osteoblast cell line (parental strain and no specific subclone), which was better suited for this study.

### 2.1. High Seeding Efficiencies after Optimization of Scaffold Pre-Treatment and Seeding

Figure 1 illustrates the used porous 3D scaffolds made from α-TCP/HA and baghdadite. The standard cylindrical α-TCP/HA scaffolds (Innotere GmbH, Germany) had a smaller total volume compared to the cubic baghdadite scaffolds. However, both scaffolds had the same porosity (50%) and pores in a comparable size range (~300 vs. ~420 µm). For α-TCP/HA, we chose a 45° shift between layers as this promised higher mechanical stability for future studies. For the same reason, the baghdadite scaffolds were cubic in geometry with the dimensions given. In both cases, we tested scaffolds with an open (all sides open) and a closed geometry (only bottom and top sides with open porosity). As in the seeding experiments, the cell suspensions remained inside the scaffolds equally well with both porosity types; we continued all experiments with the open porosity variants because these promised a better supply of cells in the interior of the scaffolds in the static culture.

MC3T3-E1 cells were used to determine suitable cell densities and incubation times for the two presented scaffold types, which proved to be critical for cell scaffold interaction and adhesion to the scaffold. A concentration of 6000 cells/µL (tested: 2500–7500 cells/µL) and an adhesion time of 3 h (tested: 45–180 min) were identified to be suitable for sufficiently fast formation of anchored cell structures from which tissues were formed. Capillary forces constituted another factor influencing seeding. Avoiding the contact of the scaffold with the bottom of the well and applying small volumes (80% of total pore volume) proved successful in terms of retaining the cell suspensions within the scaffolds until stable adhesion could occur. Homogeneous distribution of the cells was ensured by applying the cell suspension on two opposite sides of the scaffold.

The elongated incubation times (3 h) before the immersion of the scaffolds in the culture medium improved cell attachment, which was assessed by seeding efficiency analysis (Figure 2). To determine and compare the seeding efficiencies on both scaffolds, the cells that adhered to the bottom of the well were counted after 24 h incubation. From these cell numbers, seeding efficiencies were determined as described in the Materials and Methods section. Mean seeding efficiencies of 97.6% ± 2.14 for α-TCP/HA (*n* = 9) and 94.2% ± 1.95 (*n* = 7) for baghdadite scaffolds were achieved. A Mann–Whitney test indicated that the seeding efficiency for α-TCP/HA (Mdn = 97.8%) was not significantly higher (*p* < 0.05; Z = 1.87) than that for the baghdadite scaffolds (Mdn = 86.3%). The applied seeding conditions achieved high seeding efficiencies with low deviations for both scaffolds. No significant loss of loosely attached or dead cells after the immersion of the scaffolds in the growth medium was detected.

### 2.2. Proliferation and Pore Colonization Differs between α-TCP/HA and Baghdadite Scaffolds

Subsequently, cell proliferation during the course of 25 days was investigated using the CCK-8 assay kit. The assumption upon which the assay was based was that cellular dehydrogenase activity is constant in proliferative cells; therefore, its measurement enables the determination of the number of proliferative cells. A calibration curve, which relates absorbance values to known cell numbers, was used for analysis. Per time point, two α-TCP/HA scaffolds and one baghdadite scaffold were used. The proliferation values for the various time points were normalized to day 4 after seeding, which resulted in a relative evaluation of cell proliferation (Figure 3). The values increased until day 14 for both scaffolds before decreasing below the starting level at day 25. At every time point, greater proliferation activity was measured for α-TCP/HA than for the baghdadite scaffolds.

In parallel to CCK-8, we also performed transmitted light microscopy to directly observe the population of the scaffold pores with cells (Figure 4). Supported by the two-sided seeding, cells homogeneously covered the scaffold inner surfaces and filled the pore volumes at least in part after 28 days of cultivation. Due to its scaffold geometry, the α-TCP/HA scaffold pores (45° layer pattern) had a rhombic shape, while the baghdadite scaffold pores had a cubic shape. Both pore shapes and their colonization with cells were observed via transillumination imaging (overview images and magnified regions). In addition, the identity of the cells was verified by DAPI staining of their nuclei (not shown).

In the α-TCP/HA scaffolds, cells became visible after 14 days of cultivation and then continuously increased in number. In contrast, in the baghdadite scaffolds, cells could be identified within a few days after seeding, but the increase in cell mass could not be monitored using transillumination imaging, which appeared to be an effect of pore geometry. Defined concentric cell arrangements formed in the pores on the scaffolds’ top surface, while a less-structured appearance was observed on the bottom surface, where the scaffolds were in contact with the bottom of the plate. The images also show a clear narrowing of the pore channels, which, to some extent, restricts media supply to the cells inside the scaffold.

The microscopic observations also revealed a long-term negative effect of the CCK-8 assay towards cell proliferation on both scaffold types. In comparison to the non-treated scaffolds, markedly decreased cell numbers within the scaffold pores were observed after 28 days (not shown).

### 2.3. Differences in Cell Organization and Arrangement in 2D and 3D Regions

To show the distribution, morphology, and arrangement of cells in the pores of α-TCP/HA and baghdadite scaffolds, cells were labeled with DAPI and PF546-phalloidin and imaged with multiphoton microscopy (Figure 5). Here, DAPI stains the nuclei (blue) and PF546-phalloidin stains the actin filaments of the cytoskeleton (red). The cells were cultured in a normal growth medium for 21 days in both cases.

No apparent differences in terms of cell morphology were observed between the α-TCP/HA and baghdadite scaffolds. Their cell arrangements (I) were also comparable. The cells appeared elongated and formed highly interconnected networks. This indicates that both scaffolds provide a favorable environment for cell attachment, proliferation, and viability. Depending on the degree of the populating pore volume, two cell arrangements appeared within the scaffolds: (1) in partially filled pores, the cells concentrically lined the pore surfaces without bridging across the free pore volume (I-a1, and in I-a2 with an upregulated actin signal). The edges of the concentric structure were enriched with stretched cells showing a high density of shape-forming and stabilizing actin filaments (see Appendix A). (2) Away from these regions, in the 3D cell layers (Ib), the cells were much more randomly oriented and less densely packed (see Appendix A). In both scaffolds, multilayers of cells with a layer thickness of around 200 µm were formed. This is important to consider, as limitations in terms of oxygen delivery to cells in a culture are important, but they constitute an underestimated and underappreciated factor that needs to be considered [46].

When focusing on the orientation of actin filaments (II), it became clear that they were randomly oriented in fully filled or in the periphery of partially filled pores (IIa). Here, the filaments spread across all directions of the multiple layers cells. In comparison, on the scaffold surfaces, the actin filaments were highly unidirectional throughout the superficial cell layers (IIb), so the cells appeared to be much more oriented and uniform. Interesting additional observations of cell distribution were possible when focusing on the scaffolds. For the α-TCP/HA scaffolds, it became obvious that the cells grew along the gap space between neighboring material struts, whereas in the baghdadite scaffolds, the cells were restricted to the cubic pore region (see scaffold structures in Figure 1).

### 2.4. Changes during Osteogenic Differentiation

The next step was to study the osteogenic differentiation of the MC3T3-E1 cells seeded in the α-TCP/HA and baghdadite scaffolds. One goal was to extract time-resolved quantitative information using the alkaline phosphatase (ALP) assay as a representative biochemical assay. The second goal was to collect detailed structural information regarding the depth of the scaffold pores in a non-invasive, label-free manner using multiphoton imaging (MPI).

#### 2.4.1. ALP Activity Increases to a Greater Degree in α-TCP/HA Than in Baghdadite Scaffolds

ALP activity was detected in lysates from cell-seeded scaffolds at 0, 7, 14, and 21 days after being switched from the normal growth medium to the osteogenic medium (Figure 6). Reference measurements were performed, with the scaffolds cultured in the normal growth medium throughout. Differences in the total protein content of the lysates were considered after determining protein concentrations using the Bradford assay. The measured absorbance values are shown at the bottom and as relative ALP activity (normalized to day 0) at the top of the figure.

Regarding the α-TCP/HA scaffolds, a large and continuous increase in relative ALP activity until day 21 (260-fold) was detected under osteogenic conditions (in the upper part of the figure), while for the baghdadite scaffolds, the increase was less distinct (87.8-fold). This indicates distinct differentiation induced in the osteogenic medium. Strikingly, while after day 7, the increase accelerated in α-TCP/HA, it appeared to slow down in the baghdadite scaffolds. Under normal growth conditions, only a very moderate increase until day 21 was observed in α-TCP/HA (1.54-fold), but a much more pronounced one was observed in the baghdadite scaffolds (11.4-fold). The ALP activity in both scaffolds can be compared. When considering the absolute ALP activity values (in the lower part of figure), the differences between the scaffolds under osteogenic conditions become even greater (~10× increased for α-TCP/HA at day 21).

#### 2.4.2. Clear Changes and Dynamic Remodeling in Collagen-I Fiber Networks in Both Scaffold Types

Collagen-I fiber formation is a major step in bone formation and the key mediator for tensile strength. Therefore, we monitored collagen-I formation and morphology during osteogenic differentiation non-invasively using multiphoton SHG microscopy at various depths within the scaffolds. To obtain an overview of the fiber networks, we constructed mosaics of larger areas at representative z-planes (Figure 7).

We detected collagen-I fiber networks in both scaffold types at the latest seven days after transferal to the osteogenic medium. At the same time point, collagen-I was also detectable under reference conditions in the α-TCP/HA but not in the baghdadite scaffolds. After three weeks in the osteogenic medium, collagen-I was present in large abundance in all scaffold types irrespective of the media conditions (I–IV). Differences were observed primarily in the network morphology. In the normal growth medium, homogeneous networks of fine structured fibers were formed with almost no bundling of fibers (II, IV). The networks formed in the osteogenic media, on the other hand, consisted of characteristic, thicker fiber bundles of variable thickness (I, III) with empty areas in between, wherein there was no SHG signal from the collagen-I fibers.

In the corresponding SHG/AF images, which also show the co-localized signal from the cells in green (Appendix A), these signals are located in the free spaces and often show a distinct orientation (I and III). In Appendix A, SHG/AF images including large-scale mosaics are shown in image collections. These show collagen-I and cells under osteogenic and normal growth conditions for the α-TCP/HA (Appendix A) and baghdadite scaffolds (Appendix A). Here, the different structures and distributions of the fiber networks are clearly confirmed, as well as their morphology in both partially populated pores with open spaces (images 1/3; both conditions in Figure 2 and Figure 3) and in completely populated pores (images 2/4; both conditions in Figure 2 and Figure 3). The distribution and morphology of cells is clearly visible by their AF signals (green) in the mosaic images, even over complete pore diameters including the scaffold backbone in the periphery (for α-TCP/HA, green signal). Strikingly, in some images, in the pore centers and in some cases in the periphery as well, there are only signals from cells and not from collagen-I. This indicates that different stages of pore colonization are present.

#### 2.4.3. Different Patterns of Hydroxyapatite Formation between Both Scaffold Types

A late sign of osteogenic differentiation is the formation of hydroxyapatite crystals by the cells. At this stage, we imaged HA formation with the same samples and at identical time points with respect to collagen-I formation (three-week cultivation in osteogenic medium) using fluorescence labeling of HA followed by the collection of 3D image stacks using multiphoton microscopy (Figure 8). Labeling was specific to HA, as no signal was visible in the α-TCP/HA or baghdadite scaffolds without cells (not shown). Sum intensity images (from 10 equidistant images over a depth of 45 µm) reveal bright and homogeneously distributed signals in both scaffolds kept in osteogenic conditions (I, III) and only weak signals in the respective reference samples kept in a normal growth medium (II, IV). This shows that tissue mineralization by HA formation was stimulated by osteogenic conditions.

HA formation appeared after two weeks of culture in the osteogenic medium. The degrees of deposition of HA on the α-TCP/HA and baghdadite scaffolds appeared comparable in quantity. However, in the α-TCP/HA scaffolds, the deposition of HA was organized in thread-like structures (I, white arrows) that were not observed on the baghdadite scaffolds (III). These were only visible in sum intensity images, not in single images within the stacks, i.e., they exclusively extended in 3D. A possible explanation for this thread-like arrangement of HA crystals is that deposition occurred along the formed collagen-I fibers. The image sequences of the z-stacks are shown in Appendix A. The images were collected at depths of up to 150–200 µm within the scaffolds, and the fluorescence signals also contain AF from cells. It is clear how HA crystals clearly surround the cells. When comparing the two scaffold types, two further observations stand out. First, the signals on the baghdadite scaffolds are structured differently (large signal spots) compared to the α-TCP/HA scaffolds (III vs. I). Second, when comparing the reference scaffolds (IV vs. II), a much stronger degree of HA formation in the baghdadite compared to α-TCP/HA scaffolds is noticeable.

## 3. Discussion

Alpha-TCP/HA and baghdadite are established biomaterials in bone tissue engineering, are cyto-compatible and biodegradable [28], and exhibit osteogenic, osteoconductive, and osteo-inductive properties. It is hardly possible to study the influence of the material as the only factor and to keep the geometrical properties of the two scaffolds (e.g., architecture with average pore size and pore shape as well as surface properties) completely constant, which is also due to the different manufacturing technologies of the two scaffold types. Nevertheless, in our opinion, the main scaffold properties are adequately comparable. Differences in scaffold size and pore structure are discussed in detail in several places.

It was critical to test various pretreatment protocols in terms of scaffold seeding, and we reached consistently high seeding efficiency for both scaffolds (with only low deviations; see Figure 2) with the chosen duration of adhesion. The essential parameters for seeding were suitable cell numbers and densities. These were required to fill the empty scaffold volume to allow for a sufficiently high number of cells to interact with the inner surfaces of the pores. Another reason is that cell proliferation decreases during the various stages of osteogenic differentiation [22]. Therefore, cultivation was started with scaffolds pre-populated with a high number of cells. Regarding cell attachment, it was critical to completely dry the pre-equilibrated scaffolds before the addition of cells to overcome leakage, so the cells were kept within the porous network by capillary forces. Early contact of the seeded scaffold with the bottom of the culture wells had to be avoided. It was also useful to keep the cell-populated constructs in non-cell-adherent culture dishes (suspension cell plates) during the later stages of the culture to reduce the number of cells that were unrecoverable due to their growth on the bottoms of the plate wells. To attain stable cell adhesion, it was critical to provide sufficient time for cell attachment. During the time of adhesion, the cells were in a small volume of the medium for a prolonged period, which may cause cell stress via nutrient deficiency. However, we did not face major delays in initial cell proliferation after adhesion. In principle, a reduction in adhesion time could be reached either by pre-coating the scaffolds with adhesive molecules or by plasma treatment, in which the latter has a hydrophilizing effect on the scaffold surfaces that might promote cell attachment.

The proliferation assay produced interesting information. The graph shows the course of relative cell proliferation over the 25 days after seeding (Figure 3). Beginning at 100%, the values increase and reach their maximum at day 14 before they begin to decrease. Initially, the number of proliferating cells increases strongly due to the available space and media supply. With increasing cell densities, however, cell division no longer increases (maximum) and instead reverses, decreasing continuously until only few cells are still proliferating as both space and nutrients become increasingly limited. Potential reasons for the differences between the α-TCP/HA and baghdadite scaffolds may be the different scaffold sizes or pore structures (or both); however, the scaffolds can also be influenced by differences in material properties (e.g., released ions). It appears clear that both the increase and the maximum values were much higher in the α-TCP/HA compared to the baghdadite scaffolds. Within the comparably large baghdadite cubes, a fraction of cells in the scaffold core may be less proliferative, which leads to these differences. To rule out effects due to size, we will assess cubic α-TCP/HA scaffolds (edge length: 8 mm) in future studies. The sample number per time point was only two for α-TCP/HA scaffolds and one baghdadite scaffold. This was due to the limited availability of scaffolds due to the many optimization experiments in the different assays with several time points and the high demand of cells and consumables for the culture and assays to be performed. The proliferation values determined for both scaffolds, however, appear consistent and valid, especially in the course of time. The significance of the assay for porous 3D scaffolds must be examined in more detail. The CCK-8 kit always determines solely the number of proliferating cells and not the total number of cells. The CCK-8 assay was validated by the manufacturer only for cells seeded in culture plates at low to medium density. In this special 2D case, the total number of cells is identical to the number of proliferating cells until higher cell densities form. This setting, however, is not comparable with the situation in the case of the 3D scaffolds, where experiments start with a defined high seeding density, after which they quickly reach high densities within the pores. Thus, one can expect that not all cells will remain in the proliferation state for prolonged time. This is quickly shown in the time course of the curves, wherein the absorbance values only initially increased sharply and later flattened and decreased again. The obvious explanation is that only a portion of the cells actively proliferates in the pore spaces of the scaffolds. While cells that initially anchor to the inner pore surfaces should eventually enter a resting state at high cell density, cells located further toward the pore center should continue to actively proliferate in order to further fill the pore volume. The prerequisite for the accurate determination of cell proliferation is that the dehydrogenase activity in proliferating cells behaves in the same way in 3D as in 2D. The alternative to cell quantification would be to detach the cells from the scaffolds at the respective time points in parallel experiments in order to count them or to determine the total protein quantity, which would be very time-consuming. Overall, it can be stated that the kinetics of the curves may be useful to either compare the cell proliferation between scaffolds fabricated from different materials (using the same scaffold geometries) or to compare the effects of different scaffold architectures or culture conditions on cell proliferation when using scaffolds made from the same material.

The microenvironment has a strong impact on cells. To directly image the cells in the defined architectures of the scaffolds, we used transmitted light microscopy. Cell attachment is dependent on the total surface area for binding inside a pore network. As already noted, the mean pore size in the α-TCP/HA scaffolds was comparable to that if the baghdadite scaffolds. The available surface area for cell adhesion is larger in the α-TCP/HA scaffolds than in the baghdadite scaffolds, since α-TCP/HA is composed of round struts (deposited by extrusion printing (Figure 1) and its respective scaffolds have significantly smaller (330 vs. ~700 µm) diameters and thus a larger surface–volume ratio. This might explain the much larger increase in relative cell proliferation in the CCK-8 assays in α-TCP/HA compared to the baghdadite scaffolds.

Studies have shown that pore shape may also have a strong impact on cell behavior [39]. Pores are formed differently in α-TCP/HA (rhombic) and baghdadite scaffolds (cubic). In addition, the architecture of the whole pore system impacts the pores’ colonization with cells. We are aware that differential cell proliferation between scaffolds may have been influenced by pore architecture and distribution, as described in [47]. The effects of different pore systems’ architectures and sizes were observed by repeatedly subjecting cell-seeded scaffolds to transillumination imaging (Figure 4).

In the baghdadite scaffolds, the pores became overgrown concentrically (Figure 4, right and Figure 5I(a1,a2). In the α-TCP/HA scaffolds, however, due to layer shifting (by 45°), the pores became only partially visible (Figure 4, left). However, when focusing on these scaffolds with a microscope, it became clear that their population was primarily determined by cells filling the spaces between struts within a layer, so the nature of pore filling was somewhat unsymmetrical. Differences in the architecture of the pore system may also impact the diffusion of nutrients, especially at large cell densities, which may limit the proliferation rate in static cultures over extended time periods (see Figure 3).

It was not possible to focus on cells at all depths of a 3D scaffold with this technique because the image resolution was only sufficient at the periphery (i.e., the lower and upper side of the scaffolds). Nevertheless, the technique was useful for a rough assessment of the size of the cell clusters formed and their comparison between scaffolds.

The cells’ location, morphology, and orientation in the pores were verified by the co-staining of the cells with a nuclear (DAPI) and an actin filament marker (PF546-phalloidin) (Figure 5). The images were recorded from scaffolds cultured in a normal growth medium when cell density was high. Multiple images were taken as z-stacks to highlight details in 3D (Appendix A). The cells’ appearances were comparable, largely irrespective of the scaffold type, and showed a fibroblastoid morphology typical of MC3T3-E1 [48]. The orientation of actin filaments was used as an indicator of cell morphology and cell orientation. Characteristic cell arrangements became visible. It became clear how the cells lined the pores and arranged themselves concentrically (figure subsection I-a1/a2). A central round opening was left open, at the edge of which the cells flattened out considerably. There, cells with densely packed actin filaments arranged in parallel, forming a robust cell assembly (see also the collagen-I fiber networks in Appendix A), whereas the cell assemblies in the periphery were much looser, with little bundling and parallel alignment of actin filaments (I-a1/a2 and b). The remaining opening appeared to ensure that the cells remained supplied by diffusive nutrient transport. In subsection II, the different arrangement of actin filaments in the 2D and 3D environments were shown. On 2D surfaces, such as at the cell–material interface (e.g., on α-TCP/HA 2D scaffold disc) or in the pore-lining cell assemblies, high cell density (determined by numbers of stained nuclei) with strong bundling and parallel alignment (II-b) was observed. On the other hand, in the 3D cell assemblies, there was correspondingly lower cell density and no pronounced, preferred direction of the filaments (II-a). Here, the actin filaments stretched into multiple directions, because the cells had more directions in which to establish cell–cell contact.

We found the changes occurring after switching to osteogenic conditions particularly interesting because these are most relevant for the formation of bone tissue. We observed both the time course of ALP activity and the morphological and structural changes resulting from the formation of collagen-I and hydroxyapatite. There have been some studies investigating ALP activity in MC3T3-E1 and osteoblast-like cells in different settings. Ni et al. (2007) [49] compared the behavior of rat osteoblast-like cells on β-TCP versus calcium silicate scaffolds and found elevated levels of ALP activity on α- and β-calcium silicate scaffolds; however, this was only the case with 2D scaffolds. Choi et al. (1996) [50] studied the expression patterns of bone-related proteins during osteogenic differentiation and detected the maximum ALP activity at day 10; however, the cells were not grown on scaffolds. Here, we determined ALP activities in MC3T3-E1 cells cultivated in 3D scaffolds (Figure 6). The values increased in both scaffold types; however, the initial increase in relative ALP activities occurred faster and was more pronounced in α-TCP/HA compared to calcium silicate-based baghdadite (contrary to the observations of Nie et al., 2007). The differences are most pronounced three weeks after the switch to an osteogenic medium. In this medium, the relative ALP activity is three-fold higher in α-TCP/HA compared to baghdadite and about ten-fold higher with respect to absolute ALP activity values. The reason for the much less pronounced increases in the baghdadite scaffolds may be due to low ALP activity in the relatively larger scaffold cores, but this remains elusive. Interestingly, while in the normal growth medium the relative ALP activity in the α-TCP/HA scaffolds increased only very moderately (factor of ~1.5) during three weeks of measurement, in the baghdadite scaffolds, it was elevated almost ten-fold. This supports the existence of a strong osteo-inductive effect of baghdadite on the cells.

Functional bone tissue requires the formation of large amounts of collagen-I fibers, which reorganize into the typical bone structure (matrix maturation) and then calcify (HA formation), leading to distinctive mechanical properties. This process of osteogenesis occurs gradually [22]. The different steps of osteogenesis can be studied using quantitative assays; however, these cannot provide information on structural aspects. However, we believe that there is great potential in recapitulating these structural properties to obtain better artificial bone tissues. We have used multiphoton imaging (MPI) to study collagen-I matrix formation, maturation, and mineralization. The cell line MC3T3-E1 that was used is known to generally tend to synthesize collagen-I networks in an osteogenic medium and independently of added ascorbic acid. [12,39]. In this study, cell-seeded α-TCP/HA and baghdadite scaffolds pre-cultivated for 4 weeks were either converted to an osteogenic medium or continued to be grown in a normal medium (reference). After 21 days, collagen-I was found in comparable amounts in both scaffolds irrespective of the culture conditions (Figure 7). This shows that osteogenic conditions are not the trigger for collagen-I formation, but that it occurs independently through tissue maturation at a sufficiently high density of cells. However, what changed markedly was the structure of the collagen networks. This could be monitored continuously, very comprehensively, and at the depth of the scaffold pores, and we think this is of great interest with respect to bone tissue engineering.

Due to their structural properties, collagen-I fibers generate an SHG-scattering signal at a wavelength of exactly half of the excitation laser (here 810 nm). The fibers imaged are almost exclusively of collagen type I [45,51,52], which provides sufficient specificity for imaging. Assessing the quantity, distribution, and morphology of collagen-I fibers is not straightforward. On the one hand, this is due to the features of the detection of the SHG signals, which generally require a minimum amount of collagen-I fibers to be present [43,44], meaning that low amounts cannot be detected. In addition, the SHG signal varies with the thickness of the collagen-I fiber bundles formed (i.e., the density of the scattering source for the generation of SHG) [53]. Consequently, thicker and more assembled fibers are disproportionately represented while thinner ones are more difficult to discern in the images. The ELISA-based detection of collagen-I fragments released during collagen-I fibrillogenesis is more sensitive, specific, and ideally suited to quantifying collagen amounts; however, it cannot provide structural information.

Osteogenic conditions appeared to trigger two processes: (1) increased fiber bundling, i.e., a higher assembly grade to allow for more and thicker fibers to be produced, and (2) stronger parallel orientation of collagen fiber bundles and cells (visualized by their AF signal). The impressions from imaging are best represented by the image mosaics (Figure 7 and Appendix A). When the scaffolds were cultured in a normal growth medium, rather fine-structured meshes with thin fibers organized in evenly distributed networks were formed (Figure 7II,IV). In contrast, when exposed to osteogenic factors, the fibers were mostly compacted into fiber bundles, resulting in free areas without signals in their vicinity (Figure 7I,III and Appendix A—osteogenic medium, image 4). In addition, there seemed to be a trend towards the formation of preferred fiber directions, i.e., an increasing proportion of bundles arranged in the same spatial orientation (reminiscent of the structure in native bone). This is also evident in Appendix A, wherein the AF from the cells is shown in addition to the SHG signals. Here, the cells likewise aligned longitudinally, despite being under normal growth conditions (see Appendix A, images 1 and 2).

The collagen-I fibers and cells arranged somewhat differently in the α-TCP/HA scaffolds compared to the baghdadite scaffolds. In the baghdadite scaffolds, a mesh of thickened fiber bundles with gaps of relatively uniform size is present, which are regularly filled with rounded cells (Appendix A, osteogenic medium, images 1/2 and 4). In the α-TCP/HA scaffolds, on the other hand, the matrix structure more closely resembles the plywood motif typical for cortical bone (Appendix A, osteogenic medium, image 2). Furthermore, in this scaffold, the cells are more likely to be found in a stretched, uniform arrangement, unlike in baghdadite. These features may indicate that the maturation grade of the tissues was somewhat lower in baghdadite than in α-TCP/HA. A possible explanation could be the different microenvironment in the scaffold pores due to the pore architectures. As already mentioned, the α-TCP/HA scaffolds exhibit a larger inner surface area within the pores’ zone of cell–material interaction than the baghdadite scaffolds. This may have an influence on the orientation of cells and collagen-I fibers, in addition to the other influences caused by differences in scaffold architecture.

The interpretation of the collagen-I networks was somewhat complicated by the fact that they obviously change dynamically. For a complete understanding, they would have to be tracked without gaps in time. For example, while collagen-I fiber arrangements were clearly visible in marginal areas with concentric tissue arrangement in the pores (Appendix A; normal medium, images 1 and 3), this was not observed in the other images of Appendix A. This variance was visible in both scaffold types irrespective of the culture conditions, which appears to indicate the occurrence of the dynamic remodeling of these areas. The cause of this partial absence of collagen-I fiber networks (no SHG, only an AF signal from cells) could be weak adaptation to poor culture conditions. The cells may react with ECM degradation and the loosening of the respective tissue, with the aim of tissue reorganization to ensure better nutrient supply. However, the observations could also show only individual sections of a normal process, in which the different steps of pore colonization up to complete filling become visible. This partial absence of collagen-I occurred also in other locations in Appendix A (osteogenic medium, image 4) and Appendix A (normal medium, images 3 and 4), which underlies that dynamic tissue-remodeling processes are involved here. In further studies, it would be interesting to conduct a systematic temporal and spatial observation of the fiber networks. This may lead to better understanding and control of the formation of artificial bone tissues.

In general, label-free MPM proved to be very well suited for the study of osteogenesis, as it can be used continuously during a culture without causing lasting damage to the tissues. In addition, it can provide (a series of) images at greater depth (depending on the scattering properties of the turbid medium) while maintaining high resolution. Furthermore, MPM can specifically detect collagen-I fibers without the requirement for invasive staining. However, as the resolution power continues to decrease at greater depths, direct viewing into the core region of larger scaffolds is no longer reasonably possible. It would be necessary to section the scaffolds and examine the tissue on according to characteristics of the slices. However, sectioning is elaborate and requires adapted methods and pretreatments (cell fixation, staining, and tissue embedding), which need to be established for each type of scaffold.

A sign of the late phase of osteogenic differentiation is the formation of hydroxyapatite crystals [20]. As a calcium phosphate, HA is dependent on preceding ALP activity. In this study, we imaged HA and collagen-I formation at day 21 after conversion to osteogenic conditions using sum intensity images from a z-series of fluorescence images that specifically image the HA (Figure 8). While little HA formation was observed under normal growth conditions, strong HA formation was observed in both scaffolds under osteogenic conditions, indicating that this was stimulating the onset of tissue mineralization.

A marked difference in the structure of the HA signals was observed. In the case of baghdadite, the signals were of different sizes and appeared partly punctate but also showed larger patches. In the α-TCP/HA scaffolds, on the other hand, only point-like signals appeared, which showed a filamentary linear arrangement. The signals are composed of the sum of ten superimposed equidistant images with 5 µm image spacing. The linear arrangement of HA signals in α-TCP/HA can be understood as filamentous and lying obliquely in the z-plane, whereas the HA appears to be associated with (bundled) collagen-I filaments lying in the spatial plane (z-plane), which is only seen in the sum intensity images. In the baghdadite, such structures were not present, suggesting a slowed or incomplete osteogenesis process. In line with this, ALP activity in baghdadite scaffolds remained significantly below that in the α-TCP/HA scaffolds under osteogenic conditions (Figure 6). The z-stacks of the individual images are shown in Appendix A for α-TCP/HA and Appendix A for baghdadite. In the images of the videos, the AF signal of the cells is still present in addition to the HA. In contrast, in an osteogenic medium, HA formation appeared to be increased in the baghdadite scaffolds compared with α-TCP/HA, albeit at very low levels. However, this observation is consistent with the strong increase in ALP activity observed at the same time point and can be explained by the osteo-inductive effect of the baghdadite scaffolds. In summary, HA formation in the 3D scaffolds could be mapped specifically and according to depth (z-plane) and is well suited to studying the distribution and morphology of HA crystals. However, how this looks in the scaffold core is only accessible via sectioning, which is the case for collagen-I formation as well. Co-localized imaging of collagen-I and HA is possible in principle, but with the limitation that collagen-I can only be imaged in a non-native state (after the fixation step necessary for HA staining), which requires special filter sets.

In this study, we compared cell growth and osteogenic differentiation in α-TCP/HA and baghdadite scaffolds with biochemical and multiphotonic imaging. In summary, both scaffolds were equally efficient in terms of promoting cell attachment, populating pores with cells, benefiting cell morphology, and promoting distribution in a normal growth medium. Cell growth reached high cell densities according to the applied proliferation assay; however, the α-TCP/HA scaffolds were favored. Under osteogenic conditions, ALP activity increased more strongly in α-TCP/HA, especially during the later phases of the culture. In addition, the ALP data supported the notion that baghdadite had an osteo-inductive effect on the cells, whereas the α-TCP/HA scaffolds did not, which appeared to be confirmed in the imaging results due to the larger degree of HA formation. In line with the results from the ALP activity assay, an increase in tissue calcification with respect to the α-TCP/HA scaffolds was also observed in the imaging results, wherein HA crystals appeared to be neatly aligned along collagen fibers. SHG imaging showed that collagen-I fibers became increasingly bundled and oriented under osteogenic conditions, which could be explained by increased matrix maturation. The collagen-I fibers arranged somewhat differently in baghdadite compared to the α-TCP/HA scaffolds. This finding may indicate that the maturation grade of the tissues was somewhat lower in baghdadite, thereby necessitating verification via quantitative methods as well as further scientific investigation.

## 4. Materials and Methods

### 4.1. Cell Culture

Mouse pre-osteoblasts (MC3T3-E1 parental cell line [12], #ACC-210, DSMZ, Braunschweig, Germany) were kindly provided by Mahshid Monavari from the Institute of Biomaterials (Friedrich-Alexander University of Erlangen-Nürnberg, Germany). The cells were established from the calvaria of an embryo/fetus C57BL/6 mouse and were described to differentiate to osteoblasts and to produce collagen-I. The cells were cultivated at 37 °C with 5% CO_2_ and 95% humidity and verified as mycoplasma-free by PCR testing. The cells were cultivated in MEM-α basal medium (w/o nucleosides, #22561-021, Life Technologies, Darmstadt, Germany). MEM (Minimum Essential Medium) was supplemented with 10% fetal bovine serum (FBS, #10270106, Life Tech), 1% penicillin/streptomycin (10,000 U/mL, #15140122, Life Tech), and 1% L-glutamine (200 mM, #A2916801, Thermo Fisher, Darmstadt, Germany). This full medium is referred to as normal growth medium. To induce osteogenic differentiation at sufficiently high cell densities, the growth medium was replaced by osteogenic medium, which, in addition to normal growth medium, was supplemented with 50 µg/mL of ascorbic acid (#A5960100G), 10 mM of β-glycerophosphate (#G9422100G), and 10 nM of dexamethasone (#265005, all from Merck, Darmstadt, Germany). Normal growth and osteogenic media were replaced every two to three days.

### 4.2. Scaffolds

Two types of scaffolds were used. The first type consisted of synthetic calcium phosphates, mainly α-tricalcium phosphate (α-TCP), and microcrystalline calcium-deficient hydroxyapatite (HA), which mimic the composition of natural bone. These scaffolds (α-TCP/HA) were manufactured by 3D extrusion printing followed by air-drying (not heat-treated) at Innotere GmbH (Radebeul, Germany, #211CC4) [54,55,56,57]. The scaffolds had a cylindrical geometry with a diameter of 9.4 mm and a height of 3.0 mm (volume: 208 mm^3^), a high level of interconnected porosity (50%), and a high specific surface area. The strand orientation changed from layer to layer in a 45° pattern (number of layers: 12, strand diameters: 330 µm, and strand gaps: 390 µm). The average pore size was ~300 µm (pore geometry: rhombic). 2D discs from α-TCP/HA were also obtained from Innotere (#211CC2).

The second type of scaffold consisted of baghdadite (a ceramic material) and was printed by stereolithography at the University of Sydney (Australia, Biomaterials and Tissue Engineering Research Unit) [29]. Briefly, baghdadite particles were mixed with a photocurable resin (Clear resin V4, Formlabs, Somerville, MA, USA) and a dispersant (Tween 20, Sigma Aldrich, Schnelldorf, Germany) with respective ratios of 65 wt%, 17.5 wt%, and 17.5 wt% to make the ceramic resin. The resin was then printed using a desktop 3D printer (Form 2 from Formlabs). The final step was sintering (1460 °C for three hours). These scaffold cubes had lateral dimensions of 8 mm, a height of 7.5 mm (volume: 480 mm^3^), and a porosity of ~50%. The strand pattern changed in a 90° pattern from layer to layer (number of layers: 7). The average pore size was ~420 µm (pore geometry: cubic).

### 4.3. Seeding and Cultivation of Cells on Scaffolds

Prior to cell seeding, the scaffolds were sterilized for 15 min in 70% isopropanol, washed three times with PBS, equilibrated in growth medium for 24 h, and air-dried completely. Cells at passage 25 were prepared for seeding of the scaffolds. The ideal cell concentration was found to be 6.25 × 10^6^ cells/mL. Regarding the α-TCP/HA scaffolds, 80 µL of the cell suspension (500,000 cells) was dropped on two opposing sides (2 × 40 µL), while 104 µL (650,000 cells) was used for baghdadite scaffolds (2 × 52 µL). Scaffolds were held with forceps to avoid contact with the bottom of the well. This procedure helped to keep the cells inside the scaffold and to increase seeding efficiency. This resulted in ~4800 cells/mm^3^ pore volume for α-TCP/HA and ~2500 cells/mm^3^ for baghdadite scaffolds. The scaffolds were then gently placed in 6-well plates, incubated for an elongated duration of 3 h to enable cell attachment, and, subsequently, gently immersed in 4 mL resp. 8 mL (α-TCP/HA and baghdadite) growth medium. After 28 days of cultivation, half of the scaffolds were transferred to osteogenic medium, while the remaining ones were kept in normal growth medium. To evaluate the seeding efficiency after 24 h, cells that adhered to the bottom of the well were detached with trypsin and counted (Neubauer Improved cell counting chamber, #717820, Brand, Wertheim, Germany). The seeding efficiency was calculated as number of cells adhering to the scaffolds (detached minus total number of cells) related to the total number of seeded cells (in %).

### 4.4. Cell Proliferation Assay

During cultivation, cell proliferation was evaluated using the WST-8-based CCK-8 colorimetric cell-counting kit (#CK04, Dojindo, Rockville, MD, USA). The principle behind the assay is based on the observation that the amount of cell-derived WST-8 formazan is directly proportional to the number of proliferative cells. Time points for detection were 4, 7, 11, 14, 18, 21, and 25 days after seeding. The α-TCP/HA and baghdadite scaffolds were fully submerged in growth medium (2 mL resp. 3 mL) in the wells of a 12-well plate. CCK-8 reagent (220 µL resp. 330 µL, which is 10% of the total volume) was added to the wells and incubated for 2 h in a cell incubator. To ensure homogenous distribution of the reagent to the cells in the scaffolds, the liquid was manually mixed every 20 min with a pipette. For measurements, 110 µL of CCK-8 containing medium was transferred to a 96-well plate, and the absorbance (λ = 450 nm) was measured after 2 h incubation using a Victor X4 multiplate reader (Perkin Elmer, Waltham, MA, USA). As reference, a standard curve was established with defined cell numbers ranging from 0 to 40,000 cells/well (96-well plate format), cultured as monolayer in 2D. The absorbance measurements were performed thrice for each cell number after 24 h of incubation in a manner identical to the scaffold samples. As the CCK-8 kit components are supposed to be non-cytotoxic, cultivation was continued after the measurements and washing with PBS.

### 4.5. ALP Activity Assay

Cells were pre-cultivated in normal growth medium for four weeks and then either continued to be cultured in the same medium (reference) or switched to osteogenic medium. To measure osteogenic differentiation by alkaline phosphatase (ALP) activity, cell-seeded scaffolds were collected on days 0, 7, 14, and 21 after the point at which they were transferred to the osteogenic medium. The scaffolds were incubated for 1 h in serum-free medium, washed three times with PBS, and incubated in 2 mL lysis buffer for 20 min on ice. The lysis buffer consisted of PBS with 0.5% Triton^®^ X-100 (#30513, Carl Roth, Karlsruhe, Germany), 25 mM of TRIS (#10601, Carl Roth), 1 mM of MgCl_2_ (#21892, Carl Roth), and 0.1 mM of ZnCl_2_ (#Z015250G, Merck, Darmstadt, Germany). To ensure full lysis, the samples were subjected to three cycles of shock freezing (liquid nitrogen) and thawing and were then kept on ice. Special attention was paid to achieving lysis of all cells in the scaffolds. ALP activity was determined using an assay (0.4 mM CSPD™ substrate with Sapphire-II enhancer, #T2210, Thermo Fisher, Darmstadt, Germany) as a measure to determine the degree of cell differentiation to osteoblasts. A total of 20 µL of the lysate was mixed with 100 µL of the reagent in a non-transparent 96-well plate (triplicate measurement). After 20 min at room temperature (RT), chemiluminescence was measured for 0.5 s (Victor X4 multiplate reader, Perkin Elmer). Lysis buffer mixed with the ALP-reagent was used as negative control. A standard curve was generated for reference. Increased protein concentrations (0 to 200 µg/mL in 20 µg/mL steps) were generated from an albumin stock solution (fraction V, 1 mg/mL, #28341, Carl Roth) by dilution in lysis buffer (triplicate measurement). To normalize ALP activity to total protein content, protein concentration was determined using the Bradford assay (5× Roti^®^Quant, #K015.2, Carl Roth) in a 96-well plate (triplicate measurement). After 5 min at RT, the absorption was measured (λ = 595 nm, 3 s).

### 4.6. Cell Preparation for Microscopy

For imaging of fluorescence-labeled samples, scaffolds were collected on day 21 after transfer to osteogenic medium. The samples were washed with PBS and fixed with Roti^®^-Histofix 4% (#P0874, Carl Roth) for 10 min at RT. After this treatment, the scaffolds were permeabilized with 0.1% Triton X-100 (in PBS) for 5 min, submerged in staining solution (1:1000 DAPI/1:40 PF546-phalloidin in PBS), and kept light-protected for 25 min at RT. Both DAPI (#PKCA70740043) and PF546-phalloidin (PromoFluor546, #PKPF546701) fluorescence dyes were from PromoCell (Heidelberg, Germany). The samples were rinsed three times with PBS for 5 min after each preparation step. Sample preparation for imaging of hydroxyapatite (HA) particle formation was similar to DAPI/PF546-phalloidin staining. After fixation, permeabilization, and washing steps, the samples were incubated in a staining solution with 1:100 OsteoImage dye (OsteoImage^TM^ Mineralization assay, #PA1503, Lonza, Basel, Switzerland) for 30 min at RT. After this step, the samples were washed with PBS three times for 5 min.

### 4.7. Multiphoton Microscopy

Multiphoton imaging was performed with a TriMScope II multiphoton microscope (LaVision BioTec, Bielefeld, Germany). The system is based on an inverted stage microscope (Nikon Eclipse Ti) equipped with a coherent femtosecond-pulsed laser source (Chameleon Vision II, Coherent, Santa Clara, CA, USA) and a maximal output power of 3500 mW. A water immersion LD C-apochromat objective (40×, NA 1.1, WD 0.62, Zeiss, Oberkochen, Germany) and photomultiplier tube detectors (#H7422-40, Hamamatsu Photonics, Hamamatsu, Japan) were used for collecting the emissions in non-descanned configuration. Recordings (snapshots, 3D image stacks, large area mosaics) were imaged with 4× line averaging and 200 Hz line frequency. Excitation of DAPI/PF546-phalloidin stained samples occurred at 775 nm. Separation of the emission signals was performed using a dichroic mirror at 560 nm. In addition, PF546 fluorescence was blocked by a 620/60 bandpass filter. For SHG imaging of collagen-I formation, laser excitation was set to 810 nm. Separation of the emission signals was ensured using a 495 nm dichroic mirror (T495lpxr). SHG signals were isolated with an ET405/20x bandpass filter and autofluorescence (AF) signals from cells (see Appendix A) with an ET525/50 bandpass filter. The individual images in large-area mosaics had an overlap area of 10%. These are sum intensity images that derived from two separate images (with a z-distance of 2.5 µm). For imaging of HA, excitation was set at 775 nm, a 560LP dichroic mirror was used, and the signal was narrowed by a 525/50 bandpass filter. Mirrors and filters were from Chroma Technology (Bellows Falls, VT, USA) and Semrock (IDEX Health & Science, Rochester, NY, USA). All images were analyzed and processed with Fiji software (NIH, Bethesda, MD, USA).

## 5. Conclusions

In this work, different techniques for optimizing and monitoring the formation of artificial bone tissue in porous 3D scaffolds were presented. Various factors affecting scaffold pretreatment and cell colonization were highlighted, and the effects of scaffold design (size, geometry, and internal architecture) on cell physiology and nutrient delivery were discussed. The biochemical assays applied were adapted to the particular characteristics of the environment within the porous scaffolds. The multiphoton-imaging technique allowed for the visualization of cell distribution, cell morphology, and cell arrangement in the tissue at greater depths, and it facilitated the non-invasive monitoring of cells and ECM (collagen-I networks, HA crystals) continuously during culture. We believe that this imaging approach is suitable to the study of changes in the progression of osteogenesis at cellular and tissue levels. This is very beneficial for a better understanding of the underlying processes and may help generate more functional and mechanically stable artificial bone tissues. Overall, the comparative tests show a mixed picture. While the α-TCP/HA scaffolds showed increased values in the assays in terms of cell proliferation and ALP activity, a possibly better supply of the scaffold interior could not be completely excluded as a cause. In the other assays and in the imaging analysis, no clear favorite could be identified, especially since baghdadite is also superior to α-TCP/HA in other areas (mechanical and degradation properties) for certain applications. Moreover, this study was not exclusively aimed at comparing scaffold types; its secondary objective was to present the development of a method for their analysis.

This study incorporates the conditions of a static culture. In a dynamic culture, the scaffold core would be much better supplied with nutrients and oxygen. This would require specially designed bioreactors that allow for the use of defined perfusion programs (e.g., strength and duration) that must be adapted to the sensitivity of the cells to the applied shear stresses. Scaffold biodegradation profiles would also become more important in this regard, as degradation is greatly accelerated under perfusion. An alternative approach to adjusting the scaffold architecture (in terms of porosity, pore size, etc.) could be to introduce channels for media transport into the scaffolds, e.g., by omitting structures during 3D printing. In addition, more advanced image analysis, e.g., of the degree of bundling and alignment of collagen-I fiber networks (possibly supported by appropriately trained artificial intelligence), would provide information for the more accurate verification of the degree of maturity and progress of osteogenesis. ELISA assays can accurately and non-invasively detect a variety of osteogenic markers, providing direct quantitative information regarding, e.g., the number of collagen-I fibers formed. However, since ELISA assays cannot provide structural information, they can only complement imaging.

## Figures and Tables

**Figure 1 ijms-24-02999-f001:**
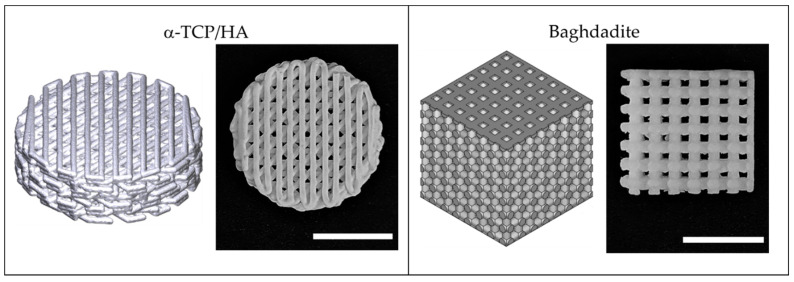
3D-printed scaffolds fabricated from α-TCP/HA calcium phosphate cement and baghdadite–calcium silicate-based ceramic used in this study. Both scaffold types are illustrated as 3D models (**left**) and photographs (**right**). Scale bar: 5 mm.

**Figure 2 ijms-24-02999-f002:**
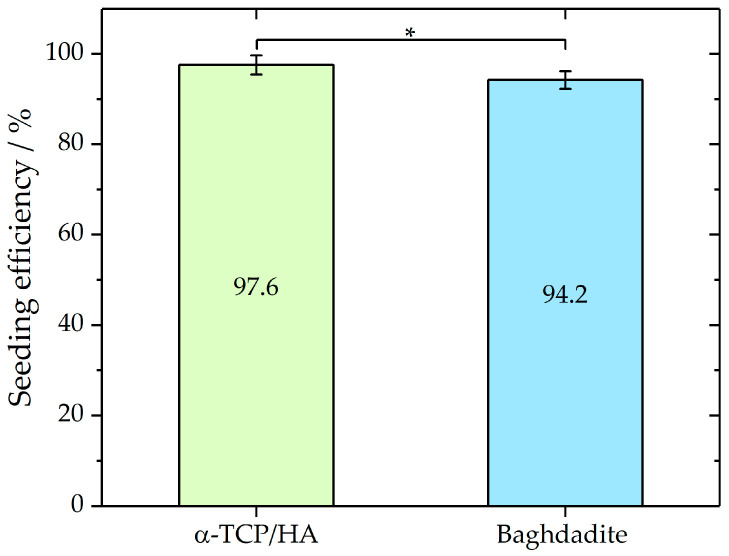
Seeding efficiencies of MC3T3-E1 in α-TCP (*n* = 9) and baghdadite (*n* = 7) scaffolds as mean with standard deviation. Mann–Whitney test *p* < 0.05; Z = 1.87. Seeding volume of the cell suspension was 80% of the total pore volume of each scaffold type. * *p* < 0.05.

**Figure 3 ijms-24-02999-f003:**
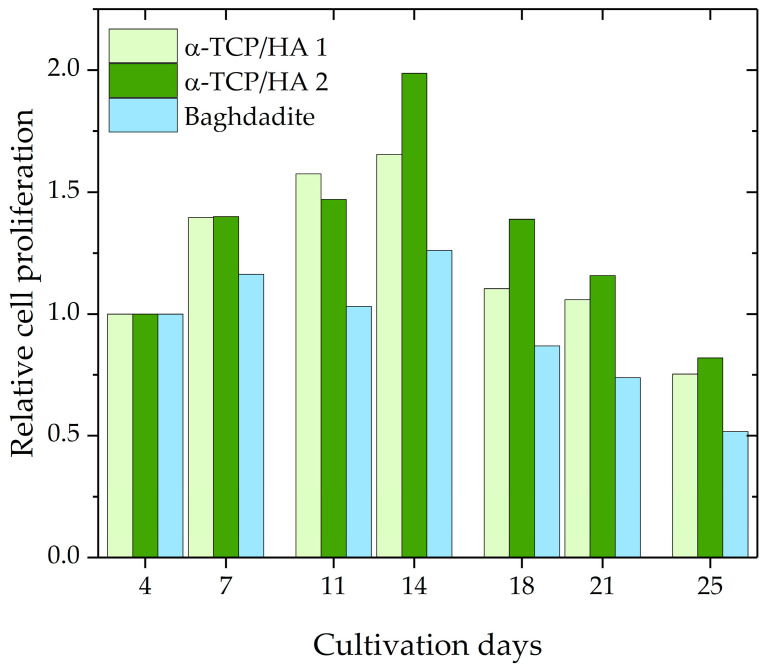
Relative cell proliferation of MC3T3-E1 cells in α-TCP/HA (n = 2) and baghdadite scaffolds (n = 1).

**Figure 4 ijms-24-02999-f004:**
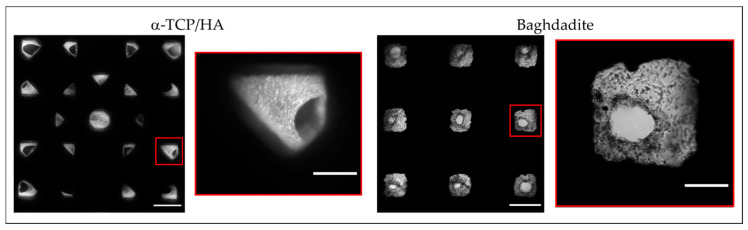
Top view transmitted light microscopy images of α-TCP/HA and baghdadite scaffold with overview images (**left**, scale bar: 500 µm) and a magnified representative pore (**right**, scale bar: 150 µm). Images were taken after 28 days of cultivation in normal growth medium. The bright regions correspond to adhered MC3T3-E1 cells within the scaffold pores (microscope: Nikon Eclipse TS100; camera: Canon EOS M10, magnification: 4×). The images are representative of five recordings per scaffold.

**Figure 5 ijms-24-02999-f005:**
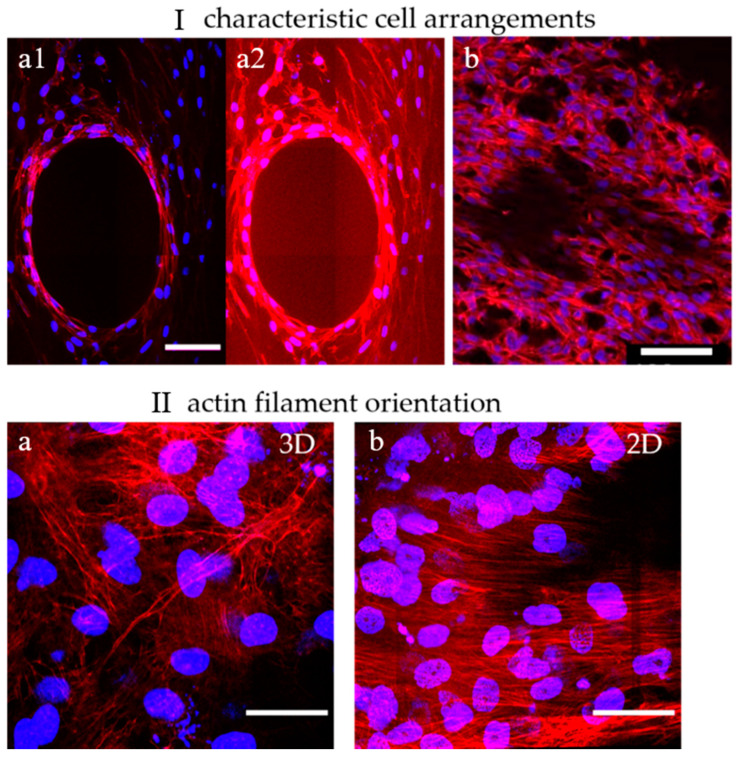
Dual-labeled MC3T3-E1 cells exemplify both cell distribution and structural features of the formed tissues within the scaffolds. (**I**) Characteristic cell arrangements in α-TCP/HA and baghdadite scaffolds. (**a1**,**a2**): The concentric arrangement of actin filament networks in the center of a partially populated scaffold pore reveals how cells are spatially ordered. (**b**): Away from that location, unconstrained spatial networks of cells were present. (**II**) Actin filament orientation in 3D and 2D regions. (**a**): In a 3D cell environment, a multidirectional orientation of actin filaments was present (α-TCP/HA and baghdadite scaffolds). (**b**) On a 2D surface (shown here: surfaces of α-TCP/HA discs), a unidirectional orientation of filaments was observed. Scale bars: (**I**): 100 µm and (**II**): 50 µm. All presented images are representative of ten replicate recordings.

**Figure 6 ijms-24-02999-f006:**
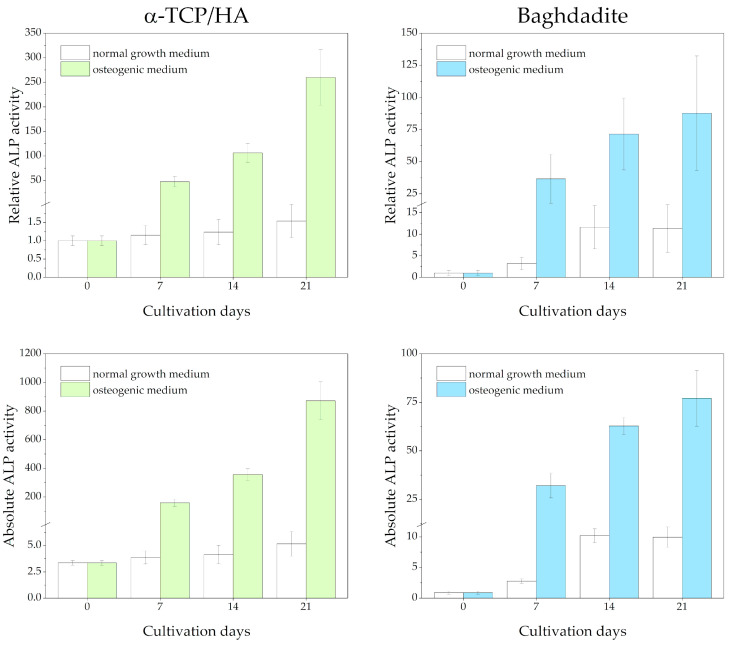
ALP activity of MC3T3-E1 cells seeded in α-TCP/HA and baghdadite scaffolds. The scaffolds were initially cultivated in normal growth medium for four weeks before transferal to osteogenic medium (colored bars) or continued to be cultured in normal growth medium (references, white bars). Three replicate experiments were performed for each time point.

**Figure 7 ijms-24-02999-f007:**
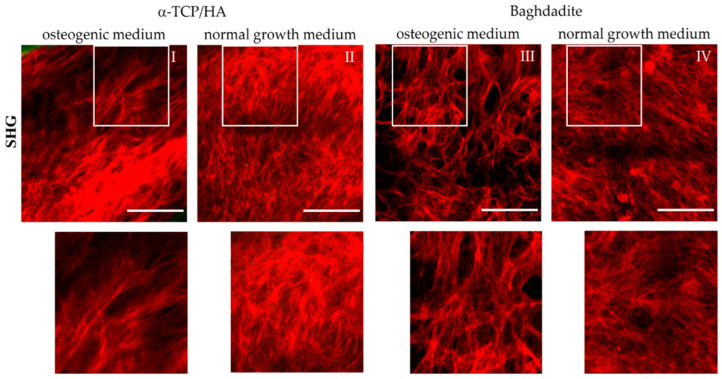
Multiphoton SHG recordings of collagen-I formed by MC3T3-E1 cells seeded in α-TCP/HA and baghdadite scaffolds. The scaffolds were initially cultivated in normal growth medium for four weeks before some scaffolds were transferred to osteogenic medium for additional three weeks (**I**,**III**) or kept for the same time in normal growth medium (references: (**II**,**IV**)). The SHG signal (red) from collagen-I fiber networks is shown, which was present at great depths of several hundred µm. The areas outlined in white are shown enlarged in the bottom row. Images are representative of at least five regions from three different scaffolds. Scale bars: 100 µm.

**Figure 8 ijms-24-02999-f008:**
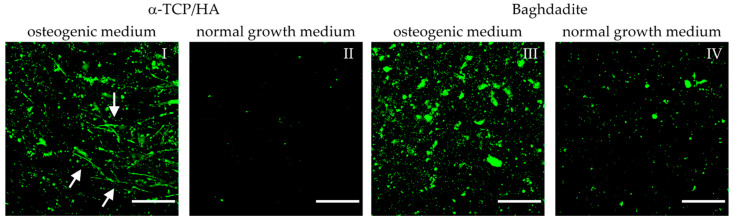
Multiphoton microscopy images of MC3T3-E1 cells seeded on α-TCP/HA and baghdadite scaffolds. Scaffolds were initially cultivated in normal growth medium for four weeks. Subsequently, transfer to and cultivation in osteogenic medium (**I**,**III**) occurred for additional three weeks (references (**II**,**IV**): scaffolds remaining in normal growth medium). Fluorescence labeling of hydroxyapatite (green) with OsteoImage mineralization assay exemplifies the mineralization of the collagen-I matrix. Images were processed from the sum intensity of a series of 10 equidistant images (∆ = 5 µm, depth: 45 µm) followed by background subtraction. Scale bars: 50 µm.

## Data Availability

The data presented in this study are available on request from the corresponding author.

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
