# Peer review of "Protocol for Cell Colonization and Comprehensive Monitoring of Osteogenic Differentiation in 3D Scaffolds Using Biochemical Assays and Multiphoton Imaging"

_ijms, 2023, doi:10.3390/ijms24032999_

Round 1
Reviewer 1 Report
In the submitted manuscript, the authors have not only demonstrated an approach for optimal cell distribution in 3D scaffolds, but also monitored osteogenic differentiation using different scientific approaches. Two types of scaffolds, Alpha-TCP/HA and Baghdadite which share similar porosity and average pore size but have different compositions and geometries are used. Various tests have been performed to analyze or optimize cell adhesion, cell proliferation and pore filling in the used scaffolds. To check osteogenic differentiation of the used pre-osteoblastic cell line within the scaffolds, ALP activity, Collagen-I fiber formation and hydroxyapatite depositions were observed. The strength of the presented work is that the authors have used multiphoton imaging to monitor cells and extracellular matrix, non-invasively during culture. The study essentially provides a set of methods for understanding cell growth, osteogenic differentiation, and tracking bone tissue development.
Overall, the manuscript is nicely written. The conclusions are substantiated by the data. Similarly, the discussion touches on some limitations of the study, related work in the field and highlights the importance of the findings. However, there are some areas where further changes might help to improve the quality of this manuscript.
· Results:
1. Throughout the results section, use different subtitles to describe findings. The subtitles may provide a brief idea about the results mentioned in the following paragraph (instead of mentioning just the used technique to conduct a particular experiment).
2. Line 184: ‘Error! Reference source not found’ is mentioned. Kindly provide a reference or clarify the bracketed words.
3. Line 288: ‘Error! Reference source not found’ is mentioned. Kindly provide a reference or clarify the bracketed words.
· Figures:
1. Figure 3: Though the trend is visible, more samples should be included in the performed experiment. It looks like ‘n’ value is just 1 here. Similarly, significance and error bars are missing in this figure.
2. Figure 5: It is not clear if the representative images (I: a1, a2, b and II: a, b) are from Alpha-TCP/HA disc or Baghdadite cube. Please mention if the cell arrangements and filament orientation are comparable in the used types of scaffolds.
· Conclusion:
1. It is not clearly stated that a particular scaffold (Alpha-TCP/HA or Baghdadite) would be effective for bone tissue development.
Author Response
Dear Reviewer 1.
Thank you for your constructive comments.
Here are my statements.
Results:
- the headings have been changed accordingly. Thank you for this good comment
- the bracketed words were deleted
- the bracketed words were deleted
Figures:
- as now noted in the Results and in the Discussion, data for alpha-TCP/HA were added. In general, the number of scaffolds was limited, especially baghdadite scaffolds (produced for us in Sydney). Since scaffolds had to be colonized, cultured and processed for the optimization experiments and the measurements with different time points (also in the other assays), more data could not be collected for the proliferation assay. However, the existing values appear plausible, especially over time, which is now also discussed in detail.
- this information is now given in the Figure legend and Results text.
Conclusion: a statement on this point can now be found in the Conclusion section.
Furthermore, based on the comments of the second reviewer, the Results and Discussion sections (especially the multiphoton imaging part) were reworded. In addition, the title was changed (now shorter), the abstract rephrased and the Conclusion section rewritten.
We hope you agree with the changes and support the publication of the manuscript.
Kind regards
Martin Vielreicher (corresponding author)

Reviewer 2 Report
Review note
This paper studies the cell contribution and osteoinductive potential of two well established biomaterials. I appreciate the authors’ works, which will help further (pre)clinical practice. However, to grow into a publication, I think there are some issues the authors need to address.
1. The novelty of this study is relatively insufficient. Like what the authors described, the two 3D scaffolds the authors used are well-studied in terms of the osteoinductive effect. The aim of this study is to provide a set of methods for tracking bone tissue development. However, using CCK-8 to test cell proliferation, using transmission microscopy to study the surface of the materials, and using ALP to test the osteogenic level are also routine methods.
2. The multiphoton imaging indeed adds up novelty and interest to this study. By using this technique, the authors provide live details of cell arrangements by both images and videos. The discussion section should be focusing on the multiphoton images and videos. These are the highlights of the study.
3. Error bar missing in figure 3.
4. Are the data from two panels in figure 6 comparable? It would be better put merge them as to better compare the two scaffolds.
5. Current discussion section is a bit too long and out of focus. I recommend the authors to focus on the results of the multiphoton imaging.
In summary, I feel the study is solid. However, improvement is needed. Particularly, the authors should highlight and exploit the novelty of this study. I hope the author(s) could find some of the above discussions helpful for improving the paper.
Author Response
Dear Reviewer 2,
thank you for your constructive comments for improving the paper.
Here are our statements.
- This point was also clear to us and we agree. Nevertheless, we used some effort to rewrite the Results and Discussion for the mentioned assays to make the findings and implications more clear.
- The discussion of MPM findings (in particular on collagen-I and HA) has now been rewritten and extended, and thereby improved.
- Figure 3: as now noted in the Results and in the Discussion sections, data for alpha-TCP/HA was added (n=2), but not for baghdadite (n=1). It is important to note that the number of scaffolds was limited in general, especially for baghdadite scaffolds (produced by the collaboration partner in Australia). Since the scaffolds had to be colonized, cultured and processed for the optimization experiments and the measurements (also in the other assays) with different time points, more proliferation assay data could not be collected by us. However, we believe that the time course of existing values are valid and plausible. This is now also discussed in more detail.
- Figure 6: the absolute ALP activity values (divided by protein concentration, but not normalized to day 0) make the data comparable. The values are much higher for alpha-TCP/HA compared to baghdadite scaffolds already in the beginning. This has now been noted and discussed. The new graph can be found in the new manuscript.
- The findings in Figures 7, 8 and S1-S3 are now discussed more in detail together with an upstream section giving more information on multiphoton imaging. The novelty of the study should be much clearer now.
Furthermore, other parts of the Results and Discussion sections were also reworded due to comments from the other reviewer. The Conclusion section was rewritten, as well as title (now shorter) and abstract reworded.
We hope you agree with the changes and support the publication of the manuscript.
Kind regards
Martin Vielreicher (corresponding author)

Round 2
Reviewer 1 Report
The authors of the manuscript titled: ‘Protocol for cell colonization and comprehensive monitoring of osteogenic differentiation in 3D scaffolds using biochemical assays and multiphoton imaging’ have made considerable changes in the revised manuscript.
In the results section, new subtitles are meaningful, which help to understand the findings. Similarly, few syntax errors are corrected. Previously missing statement about the favorability of either of the 3D printed scaffolds has been addressed in the revised conclusion section of the manuscript. Authors have clarified that the study was not exclusively aimed at comparing scaffold types, but also to present a method development for their analysis.
I am happy to see that more information has been provided for Figure 5 in the legends and results section. In Figure 3, the authors have added more data to the alpha-TCP/HA group (n=2). However, for baghdadite scaffolds, n=1 (should be 2-3 at least) is used due to the limited availability of the scaffolds which were produced in Sydney for the study group. I have no further suggestions for improvement.
Reviewer 2 Report
The authors have addressed my concerns and made significant improvements in the new version.